# Biological Control of Hyalomma Ticks in Cattle by Fungal Isolates

**DOI:** 10.3390/vetsci10120684

**Published:** 2023-12-01

**Authors:** Mohammad Ahmad Wadaan, Baharullah Khattak, Aneela Riaz, Mubbashir Hussain, Muhammad Jamil Khan, Fozia Fozia, Anisa Iftikhar, Ijaz Ahmad, Muhammad Farooq Khan, Almohannad Baabbad

**Affiliations:** 1Zoology Department, College of Science, King Saud University, P.O. Box 2455, Riyadh 11451, Saudi Arabia; 2Department of Microbiology, Kohat University of Science and Technology, Kohat 26000, Pakistan; 3Department of Animal Sciences, Facility of Biological Sciences, Quaid-I-Azam University, Islamabad 45320, Pakistan; 4Department of Biochemistry, KMU Institute of Dental Sciences, Kohat 26000, Pakistan; 5Department of Biology, Clarkson University, Potsdam, NY 13699, USA; 6Department of Chemistry, Kohat University of Sciences & Technology, Kohat 26000, Pakistan; dr.ijaz@kust.edu.pk; 7College of Professional Studies, Northeastern University, Boston, MA 02115, USA

**Keywords:** ectoparasites, *Hyalomma* spp., adult, larvae, dairy farm soil, fungal spore-free filtrate

## Abstract

**Simple Summary:**

Hyalomma ticks severely impact cattle production in Pakistan; they are responsible for transmitting the pathogens Babesiosis and Theileriosis. This study tested biocontrol of Hyalomma ticks using spore-free fungal culture filtrates from *Alternaria* sp., Aspergillus, and Penicillium isolated from dairy farm soil in Kohat. Different filtrate concentrations were applied to tick adults and larvae. No mortality occurred in the first 3 days at any concentration. At 100% concentration, fungal filtrates induced 100% mortality in adults and larvae. Decreasing the concentration lowered mortality. The effect was time- and dose-dependent, with higher concentrations causing greater mortality. Fungal culture filtrates provide biocontrol of *Hyalomma* spp. in a concentration- and time-dependent manner. Further research should explore active compounds and optimal application. This natural method could reduce chemical pesticide use against ticks and associated cattle diseases in Pakistan.

**Abstract:**

Ticks pose a major threat to cattle health and production in Pakistan because they transmit pathogens of diseases like Babesiosis and Theileriosis. *Hyalomma* spp., found across Africa, Asia, and Europe, are especially problematic. This study explored biocontrol of *Hyalomma* spp. using spore-free fungal culture filtrates collected from dairy farm soil in Kohat, Pakistan. Three fungal species of the genera Alternaria, Aspergillus, and Penicillium were isolated, and their filtrates were tested against tick adults and larvae. Filtrate concentrations were prepared at different strengths. Data were taken after the exposure of adults and larvae ticks to various concentrations of the fungal filtrates. Results indicated that at 100% concentration, all fungal filtrates induced 100% mortality in adults and larvae. Decreasing filtrate concentration lowered tick mortality. The lowest concentration caused the least mortality. The effect was time- and dose-dependent. In conclusion, spore-free fungal culture filtrates can provide biocontrol of *Hyalomma* spp. in a time- and concentration-dependent manner. Further research should explore the active compounds causing mortality and optimal application methods. The process outlined here provides a natural biocontrol alternative to chemical pesticides to reduce tick infestations and associated cattle diseases in Pakistan.

## 1. Introduction

Ticks (Acari: Ixodoidea) are blood-sucking parasites that cause major losses in animal output. Reports show that ticks and tick-borne diseases are a persistent economic danger to the livestock business [1]. Their infestations have been linked to weight loss, changes in appetite, body condition, blood composition, and decreased milk production. They can also cause hide damage and open wounds that are susceptible to secondary infections. Hyalomma is an important tick genus with a diverse host range and geographical distribution. Hyalomma, also known as hard ticks, is the largest tick family [2]. It is a member of the phylum Arthropoda, class Arachnida, subclass Acari, order Ixodida, family Ixodidae, and genus Hyalomma. This genus contains multiple species, including *H. anatolicum*, *H. aegyptium*, *H. dermacentor*, and *H. anderson*, which are usually seen on mammals’ legs, udder, tail, or perianal region. Gharbi et al. [3] classifies them as two- or three-host ticks. The multi-host tick *H. anatolicum* serves as a vector for several *Theileria* species, including *T. lestoquardi*, *T. annulata*, and *T. buffeli,* infecting big and small ruminants worldwide [4]. *Hyalomma* spp. have been found to transmit a variety of human and animal infections in Eurasia and Africa [5]. Human cases of facial paralysis caused by *H. marginatum* bites have been described, while tick paralysis caused by *Hyalomma* spp. has also been reported [6]. *Theileria annulata*, carried by *Hyalomma* spp., causes bovine tropical theileriosis, a disease that threatens approximately 250 million cattle [7].

*Hyalomma* spp. is a prominent vector in Pakistan’s livestock population [8]. These ticks not only inflict direct harm on the host animals, but they also act as vectors for many infections, resulting in economic losses in the cattle business. Pakistan’s tropical environment provides ideal circumstances for tick growth and reproduction. There are several tick species and genera in Pakistan [9]. Tick-borne diseases such as piroplasmosis and anaplasmosis are common in animals and spread by ticks in Pakistan [10]. Durrani, and Kamal [8] found the highest prevalence of *Hyalomma* spp. (15%), followed by *Boophilus* spp. (12%), *Haemaphysalis* spp. (5%), and *Rhipicephalus* spp. (3%), in the Kasur district of Pakistan. The tick’s management has generally relied on the use of conventional acaricides applied via spraying, dipping, or pour-on procedures, which are deemed successful [11]. However, in modern times, an integrated tick control management approach is acknowledged as the best answer for the livestock sector’s long-term development [12]. Chemical acaricides continue to be the dominant approach for tick management and disease prevention [13]. It is advised that they be used to minimize tick populations on afflicted cattle because they pose minimal dangers to the animals and the environment. However, acaricides can have side effects such as residue buildup in milk and meat and the development of resistance in tick populations [14].

As an alternative, the use of entomopathogenic fungi (EPFs) in tick control has attracted interest, particularly to address chemical resistance [15]. Because of their extensive distribution, diverse host range, and capacity to pierce the tick’s cuticle, fungi are known to be significant tick pathogens [16]. Several fungal species, such as Aspergillus, Beauveria, Fusarium, Paecilomyces, and Verticillium, have been linked to ticks around the world [17]. *Metarhizium anisopliae* and *Beauveria bassiana* are two EPFs that have been widely investigated and have proven efficiency against terrestrial insects, including ticks [18,19]. Similarly, several fungal and bacterial strains can also be used as biocontrol agents for different ecto- and endoparasites including *Haemonchus contortus* [20].

Ticks of the genus Hyalomma are dangerous ectoparasites that feed on animals, notably cattle. To reduce the harmful impact of *Hyalomma* spp. on animal health and productivity, effective control techniques are required. Traditional tick management treatments, such as chemical acaricides, have limits due to resistance development and environmental issues. As a result, alternative tactics such as biological management with fungal isolates have gained traction in recent years. A previous study showed that entomopathogenic fungi can be used to reduce ticks. *Metarhizium anisopliae* and *Beauveria bassiana* showed potential in tick management. Gindin et al. [21] for example, investigated the efficacy of *M. acridum* against the tick *H. marginatum* and discovered substantial fatality rates in laboratory and field testing. Similarly, Sun et al. [22] studied the pathogenicity of *Beauveria bassiana* against the tick *Rhipicephalus microplus* and found that all tick stages were fatal. The current study is notable for its emphasis on spore-free cultures recovered from the isolated fungi. Spore-free cultures offer an advantage over spore-based formulations in that they eliminate environmental concerns associated with spore dispersal while retaining fungus bioactivity [23].

The present study was initiated with the objective to isolate and identify fungal species from dairy farm soil in District Karak, Pakistan, and to evaluate the biocontrol potential of isolated fungal species against *Hyalomma* spp. in cattle. The study seeks to provide useful insights into the potential of these spore-free cultures as effective and ecologically acceptable biocontrol agents against *Hyalomma* spp. by measuring their biosynthetic activity. Understanding fungal isolate’s biosynthetic activity and efficacy against different stages of *Hyalomma* spp. is critical for creating targeted and long-term tick control tactics. Adopting fungal biocontrol agents may lessen reliance on chemical acaricides, overcome resistance difficulties, and minimize environmental. The study’s findings will help to develop safe, effective, and long-term techniques for managing *Hyalomma* spp. infestations in cattle, thereby enhancing livestock health and productivity.

## 2. Materials and Methods

The current study was conducted in the Department of Microbiology, University of Science and Technology, Kohat, Pakistan. The researchers made certain that all procedures and processes in the study were properly labeled and carried out in sterile settings. Maintaining sterile conditions is critical to ensuring the correctness and dependability of the experimental results and minimizing any potential contamination or influence. The researchers attempted to assure the validity and reproducibility of their findings by following strict sterile methods.

### 2.1. Soil Sample Collection

Soil samples were collected from 10 randomly selected places of the Zohaib dairy farm and Banda Daud Shah dairy farm in District Karak, Pakistan, during the months of July and August. The samples were pooled into a single sample weighing about 2 kg and brought to the microbiology laboratory of Kohat University of Science and Technology, Kohat, for further processing. Samples were stored at 4 °C until further use.

### 2.2. Tick Collection

Partially or fully engorged adult *Hyalomma* spp. were collected from the cattle using fine forceps stored at room temperature and identified using a given key [24]. Then, the ticks were brought to the lab to check for anti-tick activity against fungal isolates. They were brought to the lab in flasks, and the flasks were covered with a net so that ticks received fresh air.

### 2.3. Isolation of Fungi from the Soil Sample

The following approach was used to isolate fungi from the soil samples. In a glass test tube, one gram of soil was mixed with 9 mL of sterilized distilled water. After that, serial dilutions were made by transferring one ml soil suspension into other tubes containing 9 mL of distilled water, yielding serial dilutions, which were labeled as D1, D2, D3, D4, and D5. Sabouraud Dextrose Agar (Oxide, UK) was prepared and sterilized at 121 °C and 15 psi for 20 min, poured into Petri plates, and one ml of each dilution was put aseptically onto the SDA plates pretreated with chloramphenicol, an antibiotic that inhibits bacterial growth. A sterile spreader was used to spread the fungus suspension evenly throughout the plate. The plates were then incubated for 5 days at a temperature of 25 °C. Various fungal colonies were seen on the SDA plates following the incubation time. The morphological traits of these colonies were distinct. Individual colonies were chosen and sub-cultured onto fresh SDA plates, then incubated at 25 °C for another 48 to 72 h. Because only the required fungal species developed and multiplied on the plates, this phase enabled the separation of pure cultures of the fungi. Pure fungal colonies were obtained after 72 h and could be used for additional investigation or research.

### 2.4. Identification of the Isolated Fungi

Following the isolation of the fungi from the soil sample, the isolates were identified. Initially, the morphological features of fungal colonies were observed. This entailed inspecting the colonies on the culture plates for size, color, texture, and overall appearance. A lactophenol cotton blue stain was applied to glass slides to aid in identification. On the slide, a little piece of the fungal culture was stained with lactophenol cotton blue. The stained slides were then inspected under light microscopes at 10× and 40× magnification. The isolated fungi’s shape and microscopic characteristics were thoroughly investigated. A key provided by Larone (1994) [25] was used to aid in the identifying procedure. This key functioned as a guide, assisting in matching observed fungi traits with known descriptions and attributes of various fungal species. The researchers were able to narrow down the identification of the isolated fungi to specific species or genera by comparing the observed morphological and microscopic features with the information supplied in the key.

### 2.5. Extraction of Spore-Free Filtrate from the Isolated Fungi

The following processes were used to extract spore-free culture filtrate from the fungi. To begin, the fungi were moved from Sabouraud Dextrose Agar (SDA) plates to flasks containing Potato Dextrose Broth (PDB) media. Each fungal isolate was cultivated individually in flasks and cultured in a shaker incubator at 25 °C in the absence of light for seven days. To encourage fungal growth and metabolite production, the flasks were shaken at a speed of 130 rpm. Following a seven-day incubation period, the fungal broth culture was centrifuged for 15 min at 4000 rpm at −4 °C. This procedure assisted in separating the fungal cells and mycelia from the growth medium. The supernatant was then carefully filtered through Whatman No. 2 paper to remove any leftover debris or big particles. The filtered fungal culture was subsequently treated by filtration through Mellex HA syringe-driven filters with 0.45 m pore size to yield spore-free fungal metabolites. This filtration stage efficiently eradicated any leftover spores in the culture, resulting in spore-free culture filtration. The goal of this procedure was to generate a pure culture free of fungal spores [26]. Each culture filtrate was injected into SDA plates again to validate the purity of the spore-free culture. These plates were then incubated for 5 days at a temperature of 25 °C. The absence of fungal development on the plates after the incubation period revealed that the spore-free culture was truly pure, as no fungal colonies were visible. This process ensured that the isolated spore-free culture was devoid of fungal contamination [26]. 

### 2.6. Preparation of Different Concentration

The different concentrations were made from the spore-free fungal culture. The 100% concentration contained pure fungal spore culture. The 75% concentration was made by mixing 25% distilled water and 75% fungal spore culture. The 50% concentration contained 50% fungal spore culture and 50% distilled water, 25.5% was made by using 74.5% distilled water and 24.5% fungal culture filtrates, while 12.25% concentration contained 87.75% distilled water and 12.25% fungal filtrates.

### 2.7. Adult Mortality Assay

The adult motility assay (AMA) was used to evaluate the effects of spore-free cultures on adult *Hyalomma* spp. obtained from cattle abdominal segments. In this test, five adult engorged female ticks were treated in triplicate to each spore-free culture, and the adult ticks were exposed to the culture filtrates at 25 °C for 24–48 h. The spore-free cultures were prepared by diluting different quantities of distilled water (100%, 75%, 50%, and 12.5%) in small Petri plates. The technique was designed to assess the acaricidal activity of the cultures by examining tick motility. If the ticks showed reduced or no motility after being exposed to the cultures, this indicated the possibility of acaricidal effects. Ticks that showed no motility were carefully removed from the plates and placed in sterile distilled water for 10 min. If the ticks recovered their motility during this time frame, they were considered alive. If the ticks showed no evidence of motility recovery, they were deemed dead [27]. This experiment yielded useful data on the possible acaricidal activity of the generated spore-free cultures against adult *Hyalomma* spp. The researchers could determine the success of the cultures in managing these ticks, which are known to cause major harm to cattle [27] by analyzing tick motility inhibition and death.

### 2.8. Larval Mortality Assay

The larval mortality assay (LMA) was used to determine the anti-larval impact of various concentrations of the isolated fungal-free spore-free culture. Except for the positive and negative control plates, concentrations of 100%, 75%, 50%, 25.5%, and 12.25% were added to each plate containing five tick larvae. The plate containing larvae and pure spore-free culture was considered to have a concentration of 100%. The negative control plate received only 1 mL of suspension and 2.5 mL of pure distilled water, whereas the positive control plate received 1 mL of suspension and 2.5 mL of acaricides. After that, the plates were incubated at room temperature for 15 days. The plates were tested every two days following treatment, and larval mobility was used to determine anti-tick activity. Larvae with no evidence of movement were carefully removed from the plates and immersed in distilled water for 10 min. If the larvae recovered their mobility within this time frame, they were termed alive. However, if the larvae showed no evidence of motility recovery, they were deemed dead. The researchers were able to determine the anti-tick activity of the different doses of the fungal-free spore-free culture by evaluating the inhibition of larval mobility and assessing larval mortality. This assay contributed to the evaluation of the cultures’ potential as a biocontrol agent against *Hyalomma* spp. by providing useful insights into their effectiveness in reducing tick larvae.

The experiment was performed in triplicate. The percent mortality of larvae was calculated as
The percent mortality (%) = P test/P total × 100

P test: number of dead ticksP total: number of dead ticks + number of live ticks

## 3. Data Analysis

Data obtained from bio-essays—adult and larval mortality in different concentrations, i.e., 100%, 75%, 50, 25.5%, and 12.25% assays—were analyzed by ONE WAY ANOVA through statistix 9, and the graphs were made in an Excel sheet. The *p*-value ˂ 0.5 was considered significant.

## 4. Results

### 4.1. Isolation and Identification of Fungi from the Soil Sample

The colony morphology of the isolated fungal species on the SDA plates was observed and described as *Aspergillus* sp., *Alternaria* sp., and *Fusarium* sp.

### 4.2. Colony Morphology of the Fungal Isolates

The colony morphologies were studied primarily to identify the isolates (Table 1). The colony morphologies of selected isolates were different, as shown in Figure 1, Figure 2 and Figure 3. The colony morphology of *Alternaria* sp. was black to olive-black or grayish and was suede-like to floccose. Microscopically, the features included branched multicellular conidia, acropetal chains, small or elongated conidiophores, and septate division. *Aspergillus* sp. colonies were green with white shading and were fast growing. They were irregular colonies with a single layer of cells. Microscopically, they were grayish to black in appearance and circular. Colonies of *Penicillium* sp. were green, fast-growing, and irregular and had whitish cottony mycelium. Microscopy of the *Fusarium* sp. showed they were branched fulminous and blushed with septate hyaline.

## 5. In Vitro Bioassays

### 5.1. Adult Tick Mortality Assays against Fungal Isolates

The adult *Hyalomma* spp. were treated with varying concentrations of isolates taken from different fungal spore-free cultures, and their mortality rates were compared to the positive and negative control groups. Table 2 shows the results of the adult tick mortality after 15 days of treatment with various concentration of the fungal culture filtrates. The filtrates had a considerable effect on tick mortality in the case of *Fusarium* sp. At 100% concentration, the maximum adult tick mortality (5.0) was obtained, followed by a 3.3 average tick mortality at 75% fungal filtrate concentration. At a concentration of 12.25% fungal filtrate, the lowest mortality of 0.3 was recorded. Similarly, while employing a 100% spore-free culture, *Alternaria* sp. showed the greatest tick mortality at 5.0, while the mortality was noted as 4.3 at a concentration of 75% spore-free culture. At a concentration of 12.25%, the lowest adult tick mortality of 0.6 was found. At 100% isolate concentration, the *Aspergillus* sp. had a maximum mortality of 4.6. However, the mortality was reduced to 3.0 at a lower concentration of 75% fungal filtrate.

### 5.2. Tick Larval Mortality Assay against Alternaria sp.

Among the several fungal spore-free cultures, *Alternaria* sp. had the greatest mortality rate of 100% when a 100% concentration of fungal spore-free culture filtrate was used against the larvae *Hyalomma* spp. The tick larval mortality rate was 46% at a concentration of 75% spore-free culture, while, at a concentration of 12.25%, the lowest larval mortality rate of 12% was recorded, as shown in Figure 4. These data show that the spore-free culture filtrate of *Alternaria* sp. has a high efficacy in respect to larval mortality in *Hyalomma* spp. It was observed that, with the increase in the concentrations of fungal filtrate, the rate of the tick larval mortality was enhanced accordingly.

### 5.3. Tick Larval Mortality Assay against Fusarium sp.

The analysis of the data on the death of *Hyalomma* spp. larvae, caused by spore-free culture filtrates of *Fusarium* sp., demonstrates that the fungal filtrate has a considerable impact on the ticks’ larval mortality (Figure 5). At a concentration of 100%, the greatest larval mortality rate of 100% was recorded. A concentration of 75% of the fungal metabolite led to 78.3% larval mortality. At a filtrate concentration of 13%, the lowest mortality rate (12.25%) of tick larvae were reported. These findings show that spore-free culture filtrates of *Fusarium* spp. have high efficacy in producing tick mortality in larvae of *Hyalomma* spp., with greater concentrations resulting in higher mortality rates.

### 5.4. Tick Larval Mortality Assay against Aspergillus sp.

In terms of the nematicidal effect of *Aspergillus* sp. against the larvae of *Hyalomma* spp., the highest fungal filtrate concentration (100%) resulted in the maximum mortality of the tick larvae. This was followed by 61% tick larval mortality at the fungal filtrate concentration of 75%. At the lowest filtrate concentration for *Aspergillus* sp., the larval mortality rate was at the lowest concentration level of 12.25%. The results shown in Figure 6 depict that the spore-free culture of *Aspergillus* sp. has substantial acaricidal activity against adult *Hyalomma* spp., with greater concentrations resulting in higher mortality rates of the tick larvae.

### 5.5. Larval Mortality Assay against the Fungal Isolates

Table 3 shows the larval mortality rates after 15 days of treatment with various doses of the culture filtrates of *Fusarium* sp., *Aspergillus* sp., and *Alternaria* sp. The results show that these fungal isolates are effective at inducing mortality in *Hyalomma* spp. larvae. The fungal isolates showed 100% larval mortality at the maximum filtrate concentration (100%). Lower fungal filtrate concentrations resulted in variable degrees of larval mortality. *Alternaria* sp. at a 75% concentration resulted in 60% larval mortality, while a 12.25% concentration resulted in the lowest larval mortality (13%). *Fusarium* sp. caused 100% larval death when its filtrate concentration was 100%. The death rate was 46% at 50% concentration, whereas the lowest concentration (6.25%) of fungal filtrate induced the lowest larval mortality (13%). *Aspergillus* sp., on the other hand, showed 100% larval mortality at 100% filtrate concentration and 46% mortality at 50% concentration. The larvae mortality rate was 20% at the lowest concentration of *Aspergillus* sp. (6.25%). These data suggest that the fungal spore-free culture filtrates from *Fusarium* sp., *Aspergillus* sp., and *Alternaria* sp. have the capacity to kill *Hyalomma* spp. larvae.

## 6. Discussion

Ticks are obligate blood-sucking ectoparasites that infest 80% of the cattle worldwide [28,29]. *Hyalomma* spp. are well-known ectoparasites that prey on animals, notably cattle. Ticks spread a variety of infections and create economic losses in the livestock business [1]. (Sajid, 2017). Traditional tick management treatments, such as chemical acaricides, have limits due to resistance development and environmental issues. Furthermore, the use of these products is challenged by the introduction of invasive species, the quick development of physiological insecticide and acaricide resistance, and their non-target effects on human health and environment, as per Cafarchia et. al. [30].

As a result, alternative control tactics, such as biocontrol utilizing fungal isolates, have risen in popularity. When compared to chemical acaricides, using fungal isolates for tick control has environmental benefits. Because of their biodegradability and low toxicity to non-target creatures, fungi are often regarded as environmentally beneficial [14,31]. Biological control by entomopathogenic fungi (EPF) is one of the most promising options for tick control (Polar et al., 2005). Furthermore, fungal isolates can survive in the environment and provide long-term tick control [17,18,19]. It has been demonstrated that the volatile toxic metabolites and a chitinase, secreted by a fungus *Galactomyces* sp., can affect the longevity of adult ticks [32].

However, additional study is needed to investigate the potential ecological implications and safety of fungal biocontrol agents to ensure their long-term use. Integration with other control strategies may be required to maximize the efficacy of fungal biocontrol agents. By targeting distinct life stages or harnessing synergistic effects, combining fungal isolates with botanical acaricides or insect growth regulators can improve tick control. A recent study indicated the successful integration of *Beauveria bassiana* with essential oils, resulting in improved tick control on cattle [33]. For instance, *M. anisopliae* greatly reduced tick infestation in cattle. These fungi use a variety of bioactive substances and processes to pierce the tick’s cuticle, resulting in internal colonization and tick mortality [34,35].

Adult *Hyalomma* spp. were exposed to spore-free fungal culture filtrates at various concentrations (100%, 75%, 50%, 25%, and 12.50%) in this study. The treatment lasted 15 days, and the survival index (SI) was calculated at regular intervals. The findings were compared to prior research on livestock, which found that isolates of *M. anisopliae* caused 100% mortality in engorged female ticks [36]. The larvae of *Hyalomma* spp. were treated for 15 days with varied concentrations of spore-free cultures in the current investigation. Following the treatment period, the effectiveness of various concentrations was assessed. The spore-free culture with a concentration of 100% was found to be highly effective in the biological control of engorged female *Hyalomma* spp.

Furthermore, it was discovered that extract concentrations of 100%, 50%, and 25% *Fusarium* sp., *Aspergillus* sp., and *Alternaria* sp. resulted in 100% larval death in *Hyalomma* spp., owing to the potency of the extracts. It is important to note that fatality rates differ amongst tick species. These findings show the utility of spore-free cultures of *Fusarium* spp., *Aspergillus* spp., and *Alternaria* sp. in the suppression of *Hyalomma* spp. larvae. More research is needed to determine the mechanisms of action and bioactive chemicals that are causing the observed mortality rates. A similar study conducted by Fernandes et al., [37] tested the virulence of 60 Beauveria-like isolates in *R. microplus* larvae and verified that larvae from different origins had different susceptibilities to the fungal isolates.

As an alternative technique, the biological management of *Hyalomma* spp. in cattle using fungal isolates has tremendous potential. Recent research has emphasized the biocontrol efficiency of entomopathogenic fungi and the relevance of knowing their mode of action and molecular interactions [38]. So, considering the environmental benefits and researching integration with other control approaches can improve the efficacy of fungal biocontrol agents. To optimize fungal biocontrol techniques and assure their practical application in tick management programs, more research and field experiments are required. More than 170 entomopathogenic fungi (EPFs)-based products are commercially available [39]. Great improvements to the commercialization of these products is required to make them competitive against chemical pesticides.

The findings in the current research show that spore-free culture filtrates of *Fusarium* sp., *Alternaria* sp., and *Aspergillus* sp. can cause considerable mortality in adult *Hyalomma* spp. in the controlled environment, and the mortality rates differed according to their filtrate concentrations. However, under natural conditions, the effect decreases due to the influence of various environmental factors, such as temperature, relative humidity, and exposure to solar radiation [27,40]. Therefore, it is necessary to evaluate the action of fungi under diverse environmental conditions.

The entomopathogenic fungi have several advantages over the chemical acaricides, such as minimal environmental pollution, low risk for animal health, and high efficacy against pests [35]. So, a possible impact of these fungal filtrates on animals and their side effects on cattle need to be evaluated. Similarly, such fungi remain viable on cattle skin for several weeks [41], so the filtrates’ application at different intervals would be an interesting protocol to explore in further studies. Further research is needed to better understand the underlying mechanisms and enhance the usage of these spore-free culture filtrates for better tick management.

## 7. Conclusions

The current study sought to assess the biosynthetic activity of spore-free filtrates of some fungal species against *Hyalomma* spp. larvae and adults. In case of adult tick mortality, *Alternaria* sp. gave a better result, while *Fusarium* sp. was found more effective in larval motility assays. Different concentrations of spore-free filtrate were used to test their efficacy in causing tick mortality at various life stages, and the mortality increased with the increase in culture filtrate concentrations. Further research is required to investigate the chemical nature of the fungal metabolites such that their mechanism of action may be evaluated. Moreover, these fungal species need molecular identification up to their species level.

## Figures and Tables

**Figure 1 vetsci-10-00684-f001:**
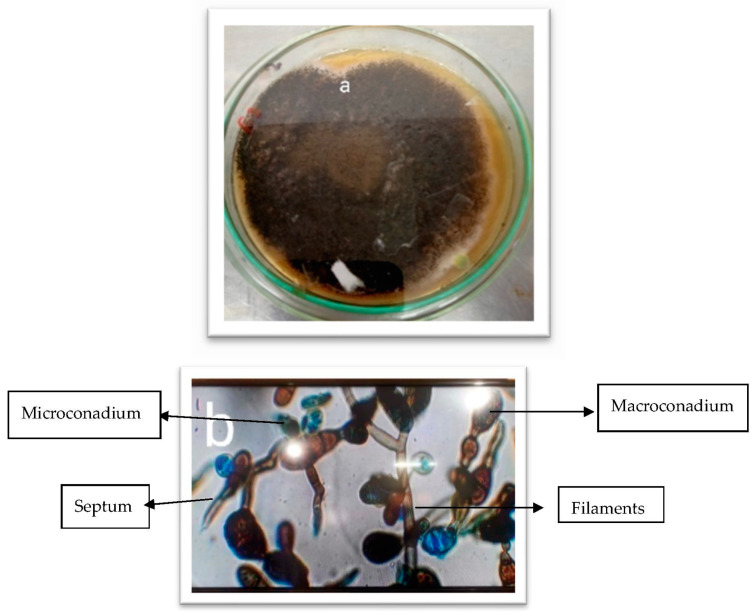
Isolation and identification of fungal isolates. (**a**) *Alternaria* sp. colony on SDA plate. (**b**). *Alternaria* sp. microscopic view at 40×.

**Figure 2 vetsci-10-00684-f002:**
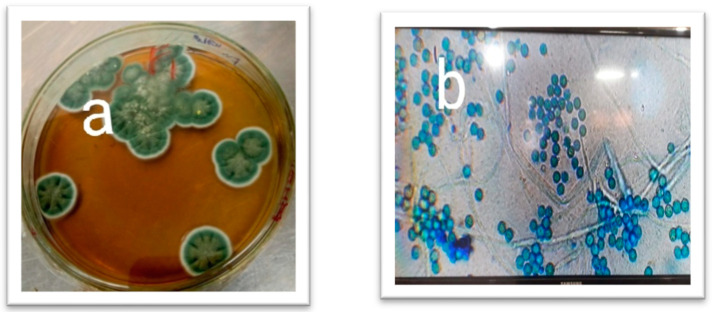
Isolation and identification of fungal isolates. (**a**) *Penicillium* sp. colony on SDA plate. (**b**). *Penicillium* sp. microscopic view at 40×.

**Figure 3 vetsci-10-00684-f003:**
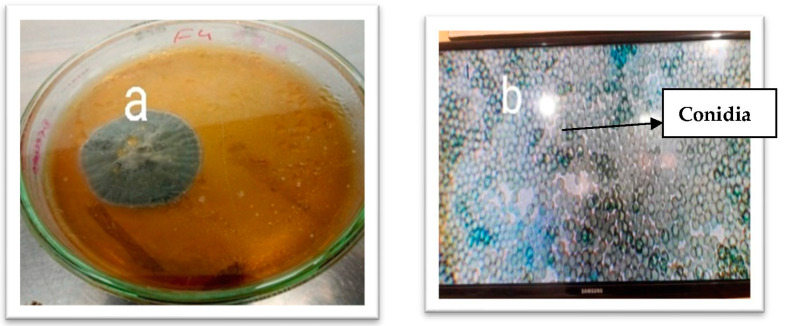
Isolation and identification of fungal isolates. (**a**) *Aspergillus* sp. colony on SDA plate. (**b**). *Aspergillus* sp. microscopic view at 40×.

**Figure 4 vetsci-10-00684-f004:**
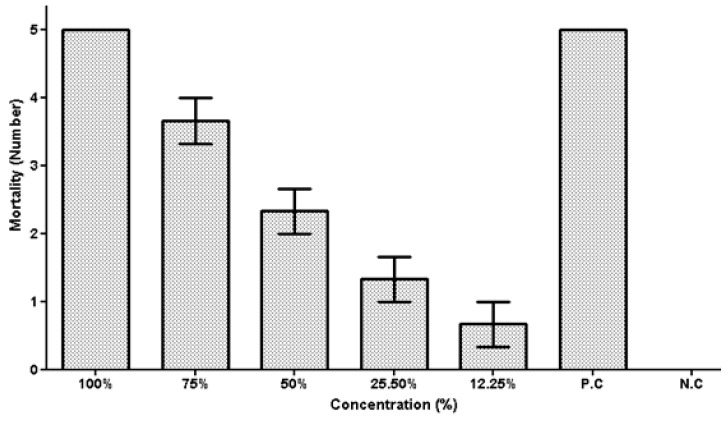
Mortality of *Hyalomma* spp. larvae after exposure to different culture filtrate concentrations of *Alternaria* sp.

**Figure 5 vetsci-10-00684-f005:**
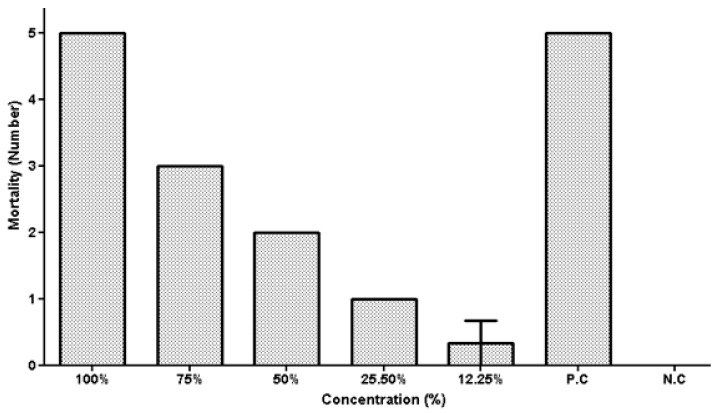
Mortality of *Hyalomma* spp. larvae after exposure to different culture filtrate concentrations of *Pinicillium* spp.

**Figure 6 vetsci-10-00684-f006:**
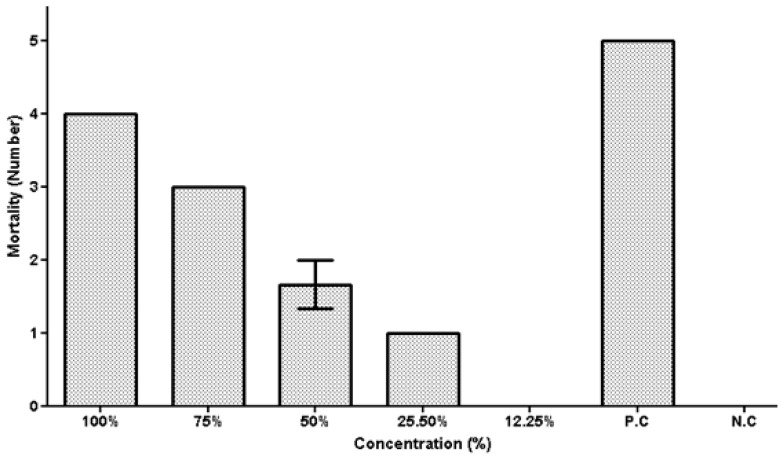
Mortality of *Hyalomma* spp. larvae after exposure to different culture filtrate concentrations of *Aspergillus* sp.

**Table 1 vetsci-10-00684-t001:** Colony morphology of the isolated fungal species.

Fungal Isolates	Colony Morphology		
	Color	Shape	Elevation
*Alternaria* sp.	Dark brown	Irregular	Septate
*Aspergillus* sp.	Greenish to black	Filamentous	Septate
*Penicillium* sp.	Gray	Circular	Septate

**Table 2 vetsci-10-00684-t002:** Adult tick mortality, as observed after 15 days of treatment with fungal culture filtrates.

Fungal Culture Filtrate Concentrations (%)	Adult Tick Mortality (n = 5)			Mean Adult Tick Mortality
	*Fusarium* sp.	*Alternaria* sp.	*Aspergillus* sp.	
100.0 (Positive control)	5.0	5.0	4.6	4.60
75.0	3.3	4.3	3.0	3.53
50.0	2.3	3.0	2.3	2.53
25.5	1.0	1.6	1.3	1.30
12.5	0.3	0.6	0.4	0.43
No Culture Filtrate (Negative Control)	0	0	0	0

*p*-value ˂ 0.05.

**Table 3 vetsci-10-00684-t003:** Tick larval mortality, as observed after 15 days of treatment with fungal culture filtrates.

Fungal Culture Filtrate Concentrations (%)	Tick Larval Mortality (n = 5)			Mean Larval Mortality
	*Fusarium* sp.	*Aspergillus* sp.	*Alternaria* sp.	
100.0 (Positive control)	5.0	5.0	4.0	4.67
75.0	3.6	3.0	3.0	3.20
50.0	2.3	2.0	1.6	1.97
25.5	1.3	1.0	1.0	1.10
12.5	0.6	0.3	-	0.30
No CF (Negative control)	0	0	0	0

*p*-value ˂ 0.05.

## Data Availability

All the relevant data is provided in the article.

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
