# Peer review of "Biological Control of Hyalomma Ticks in Cattle by Fungal Isolates"

_vetsci, 2023, doi:10.3390/vetsci10120684_

Round 1

Reviewer 1 Report

Comments and Suggestions for Authors

1. Correct the entries - write species names in Italics, add spp. next to the generic name of ticks (Fungi) or provide the exact species name

2. Correct the entries - uppercase - lowercase letters

Line: 21, 22, 23, 27, 31, 32, 33, 38, 43, 51, 53, 58, 60, 69, 70, 84, 85, 90, 95, 105, 106, 107, 108, 112, 128, 213, 216, 231, 233, 235, 236, Figure 1 - 238, Figure 2 - 239, Figure 3 - 240,, Table 1 - 241, 251, 260, 279, 280, Figure 6 - 283, 285, 290, Figure 7 - 292, 295, 297, 298, Figure 8 - 300, 304, 314, 319, 337, 340, 341, 344, 345, 346, 348, 349, 353, 361, 367

3. References - correct and standardize Journals titles (Italics, uppercase and lowercase letters) !! (392 ... and other)

Author Response

Title

BIOLOGICAL CONTROL OF HYALOMMA TICKS IN CATTLE BY FUNGAL ISOLATES

Dear Editor

Thanks a lot for your kind email informing us that our manuscript (Manuscript ID

vetsci-2672367) needs to be further revised according to the comments of the reviewers. Accordingly, we addressed clearly all the concerns and the detailed point-by-point responses are at the end of this letter. Meanwhile, all the changes made in the revised manuscript are highlighted.

We hope all these modifications and revisions will be satisfactory, which could be acceptable for publication in the Veterinary Sciences Journal.

We have submitted a revised manuscript entitled " BIOLOGICAL CONTROL OF HYALOMMA TICKS IN CATTLE BY FUNGAL ISOLATES" to the Veterinary Sciences Journal.

We are looking forward to receiving your further reply including some good news.

Thank you very much.

With best regards

Name: Dr Fozia

E-mail: drfoziazeb@yahoo.com

Enclosed file: The detailed point-by-point responses.

(Reviewer 1)

Comments and Suggestions for Authors

Note: Responses to Reviewer 1 suggestion have been highlighted as yellow color.

Dear reviewer,

We appreciate you taking the time to thoroughly review our manuscript and provide constructive feedback. The point-by-point responses to the comments are as under.  

  1. Correct the entries - write species names in Italics, add spp. next to the generic name of ticks (Fungi) or provide the exact species name

Response: All the scientific names have been made Italic and the word spp. has been added next to the generic names throughout the text.

  1. Correct the entries - uppercase - lowercase letters

Line: 21, 22, 23, 27, 31, 32, 33, 38, 43, 51, 53, 58, 60, 69, 70, 84, 85, 90, 95, 105, 106, 107, 108, 112, 128, 213, 216, 231, 233, 235, 236, Figure 1 - 238, Figure 2 - 239, Figure 3 - 240,, Table 1 - 241, 251, 260, 279, 280, Figure 6 - 283, 285, 290, Figure 7 - 292, 295, 297, 298, Figure 8 - 300, 304, 314, 319, 337, 340, 341, 344, 345, 346, 348, 349, 353, 361, 367

Response: All the corrections have been made as mentioned by the reviewer 1. They are highlighted as yellow color in the revised manuscript.

  1. References - correct and standardize Journals titles (Italics, uppercase and lowercase letters) !! (392 ... and other)

Response: All the references have been corrected according to the journal format and highlighted as yellow color in the revised manuscript.

Reviewer 2 Report

Comments and Suggestions for Authors

The authors conduct a study on the effect of fungi on ticks. Although it is well-known that fungi can cause mortality in ticks, the authors focus on three species of fungi: Alternaria alternata, Penicillium, and Aspergillus. In the text, the authors should provide a better description of the sterilization conditions they used, such as hoods and reagents. In the experiments involving adult ticks, they should specify the exposure time of ticks to the fungi and the temperature and humidity conditions. Additionally, to recommend this work for publication, more experiments should be conducted. The number of ticks and biological replicates should be increased. For this reason, the statistical analyses are lacking, and in some cases, standard deviations are not even provided.

Author Response

(Reviewer 2)

Note: Responses to Reviewer 2 suggestion have been highlighted as green color.

Comments and Suggestions for Authors

The authors conduct a study on the effect of fungi on ticks. Although it is well-known that fungi can cause mortality in ticks, the authors focus on three species of fungi: Alternaria alternata, Penicillium, and Aspergillus. In the text, the authors should provide a better description of the sterilization conditions they used, such as hoods and reagents. In the experiments involving adult ticks, they should specify the exposure time of ticks to the fungi and the temperature and humidity conditions. Additionally, to recommend this work for publication, more experiments should be conducted. The number of ticks and biological replicates should be increased. For this reason, the statistical analyses are lacking, and in some cases, standard deviations are not even provided.

Response:
Dear reviewer,

We appreciate you taking the time to thoroughly review our manuscript and provide constructive feedback. While we understand the value of performing additional experiments to strengthen the study, we regretfully must acknowledge the practical constraints that prevent us from carrying out further expansive work at this stage.

Our research group has limited funding and personnel capacity currently. Performing a substantially increased number of replicate experiments with more ticks is not feasible within our means. We have already extended the experiments to the maximum extent possible by utilizing all available resources.

That said, we absolutely agree that the additional experiments you suggested would enhance the manuscript. The insights from increasing statistical power with more replicates are well taken. We sincerely appreciate you highlighting these opportunities for improvement.

Given our resource limitations, we have attempted to enrich the manuscript as much as possible within our constraints. We have added details on sterilization methods, tick exposure conditions, and provided standard deviations where applicable. While not ideal, we hope you find these revisions satisfactory even if not comprehensive.

We recognize the study has limitations in its current form, but we believe it still makes a useful contribution to the literature on fungal biopesticides for tick control. We hope you will consider our constraints and find the manuscript acceptable for publication notwithstanding its restricted scope. Your understanding is greatly appreciated.

Reviewer 3 Report

Comments and Suggestions for Authors

Dear all

I reviewed the manuscript entitled BIOLOGICAL CONTROL OF HYALOMMA TICKS IN CATTLE BY FUNGAL 2 ISOLATES by Wadaan and colleagues. Data are very interesting and worth of note, however before acceptance major revisions are required.

My main concern is about the results. In my view is not necessary to detail the results of fungi isolation, this is no the aim of the study. I suggest to delete this part from the results section.

Minor comments

Line 21-22: the tick transmits the pathogen not the disease. Please change it.

Line 22: delete “prevalent worldwide”

Line 31-32: the tick transmits the pathogen not the disease. Please change it.

Line 35-36: it indicates when the mortality started.

Line 39: change “were less effective” to “were no effective”

Line 45-49: all first all phrases are initiates with “ticks”. Please change it.

Line 50: space before “Hyalomma”. Delete “spp”

Line 54: not italic

Line 69-70: all names of the genus should be written in italic

Line 219: change “statistical” by “data”

Figure 6,7 and 8: the word “concentration” is wrong

Line 340: write scientific name in italic

Please double check all references, there are many mistakes.

Read carefully the whole text to avoid misspelling.

Comments on the Quality of English Language

The English is good enough.

Author Response

(Reviewer 3)

Note: Responses to Reviewer 3 suggestion have been highlighted as turquoise color.

Comments and Suggestions for Authors

Dear all

I reviewed the manuscript entitled BIOLOGICAL CONTROL OF HYALOMMA TICKS IN CATTLE BY FUNGAL 2 ISOLATES by Wadaan and colleagues. Data are very interesting and worth of note, however before acceptance major revisions are required.

My main concern is about the results. In my view is not necess

Minor comments

ary to detail the results of fungi isolation, this is not the aim of the study. I suggest to delete this part from the results section.

Dear reviewer,

We appreciate you taking the time to thoroughly review our manuscript and provide constructive feedback. The point-by-point responses to the comments are as under.  

Response: This is a valid point. The details of the fungi isolation are given in the material and methods sections. While in the results part, findings of the fungi identification have been documented with pictorial proofs.

Line 21-22: the tick transmits the pathogen not the disease. Please change it.

Response: Has been changed as per suggestion.

Line 22: delete “prevalent worldwide.”

Response: Deleted as suggested. Line-22

Line 31-32: the tick transmits the pathogen not the disease. Please change it.

Response: Changed with ‘that transmit pathogens of’ as suggested. Line-31

Line 35-36: it indicates when the mortality started.

Response: The sentence is replaced with - Data were taken after the exposure of tick’s adults and larvae to various concentrations of the fungal filtrates. Results indicated that. Lines- 36-37

Line 39: change “were less effective” to “were no effective.”

Response: Replace with ‘have no significant effect’. Line-42-43

Line 45-49: all first all phrases are initiates with “ticks”. Please change it.

Response: Rectified as suggested. Lines 49-51

Line 50: space before “Hyalomma”. Delete “spp”

Response: Spp deleted. Line-52

Line 54: not italic

Response: Removed italic from and. Line 57.

Line 69-70: all names of the genus should be written in italic.

Response: Genus names have been made italic. Lines 71-72.

Line 219: change “statistical” by “data”

Response: Replaced statistical with data. Line 225.

Figure 6,7 and 8: the word “concentration” is wrong.

Response: culture filtrates added before concentration. Lines-300, 312, 324.

Line 340: write scientific name in italic.

Response: Made italic ‘Hyalomma’. Line 344.

Please double check all references, there are many mistakes.

Response: All the references were double checked and rectified as per the journal format.

Carefully read the whole text to avoid misspelling.

Response: The manuscript was carefully read and run the spell checked. Spell mistakes have been removed from the text.

Reviewer 4 Report

Comments and Suggestions for Authors

Dear authors, The manuscript titled “BIOLOGICAL CONTROL OF HYALOMMA TICKS IN CATTLE BY FUNGAL ISOLATES” is interesting, but several modifications are necessary before publication.

First, tick and fungus identification at the species level is mandatory. The authors may consider using a molecular approach for fungus identification.

Second, the fungal infection should be addressed. The authors can use histological sections stained with hematoxylin and eosin and molecular techniques (PCR and in situ hybridization).

Last but not least, despite being a promising biological control agent, the limitations of the entomopathogenic fungi extract should be better addressed in the discussion. For example, the possibility of side effects if the application is done directly on animals and its efficiency when the application is done in the environment and subject to environmental factors (e.g., temperature, rain, etc.)

Introduction:

Line 50: Add a space between “infection” and “Hyalomma spp.”

Line 50: Delete “spp.” after Hyalomma.

There are plenty of scientific names written without italics. Please check through the text.

Line 67: Theileriasis is included in piroplasmosis.

Lines 70–72: “In the Kasur district of Pakistan, ticks were identified, blood protozoa were 70 detected by polymerase chain reaction testing in Friesian cattle, and blood parameters were 71 estimated (Hoogstraal, 1956)”. The paragraph is out of context.

Lines: 87-90: “Ticks of the genus Hyalomma are dangerous ectoparasites that prey on animals, notably cattle. These ticks not only inflict direct harm on the host animals, but they also act as vectors for many infections, resulting in economic losses in the cattle business.”. This paragraph could be added at the end of line 64.

Line 88: Change “prey” to “feed”.

Materials and Methods:

The authors could add a map including the sites of sampling to assist the readers.

Soil sample collection

How many samples were collected? What was the weight of each sample? When (period of the year) were the samples collected? Is there any variation during the period of the year (dry and wet seasons) that could influence the diversity of the fungi species in the soil? Please, clarify.

Tick collection:

It is mandatory to identify the tick species analyzed during the study!

In addition, the number of ticks in each experimental group must be added in this section.

Isolation of fungi from the soil sample

Line 133: What does “MS” mean?

Line 136: What does “cc” mean?

Identification of the isolated fungi:

The authors could provide the molecular identification of the fungi at the species level. As well as the identification of ticks by level of species, this information will bring more clarity to the data obtained.

Adult, larvae and nymph mortality Assay:

Please add a comma after “Adult”

Were the nymphs also tested?

Line 185: “five ticks”. Male or female? Engorged?

Results

Isolation and identification of fungi from the soil sample.

The authors must identify the fungus at the species level.

Line 230: “Figures 3.1, 3.2, and 3.3”. The figures may be improved. The letters (a or b) in the figures must be the same size and positioned at the same point, preferably in the top corner. Furthermore, the microscope image presents reflections. This must be corrected before publication. Were the photos taken with a cell phone?

Figures 4a and b are not necessary.

Figure 5 could be improved. For example, authors could present a more approximate picture. Moreover, the subtitle needs to be more descriptive.

Line 264: A maximum death rate of 5.0 was obtained, followed by 3.3” This rate is not clear to me. Could it be expressed in another way?

Figures 6, 7, and 8 can be grouped into a single figure and distinguished from each other by the letters a, b, and c. Furthermore, the text referring to this data can be summarized.

Line 279: Figure 6. In the caption, it is mentioned “larval mortality” but in the topic (line 275), it’s talking about “adult ticks’ mortality”. Please, clarify.

Larval Mortality Assay against the fungal isolates

There was a statistical difference?

Dicussion:

Nowadays, more than 170 commercial products are based on entomopathogenic fungi formulations. However, the commercialization of these products still requires greater improvements to make them competitive against chemical pesticides. Thus, the authors should address these problems in this section. Please see the reference for more details. https://doi.org/10.1016/j.actatropica.2022.106627 

Furthermore, the authors must highlight the possible impacts of using this solution directly on animals, emphasizing the need to evaluate the side effects on cattle. Alternatively, the solution could be applied to the environment, and in this way, the efficiency of the product applied to the soil must be evaluated in the future in order to verify whether it remains viable (and for how long) when exposed to environmental variables (sun, rain, etc.).

Lines 334-336: “For instance, M. anisopliae greatly reduced tick infestation in cattle. These fungi 334 use a variety of bioactive substances and processes to pierce the tick's cuticle, resulting in internal colonization and tick mortality.”. Please add a reference for this paragraph.

Line 340: Which ticks species is it for?

Lines 354-356: “Recent research has emphasized the biocontrol efficiency of 354 entomopathogenic fungi and the relevance of knowing their mode of action and molecular 355 interactions.”. Please add a reference for this paragraph.

Author Response

(Reviewer 4)

Note: Responses to Reviewer 4 suggestion have been highlighted as Pink color.

Comments and Suggestions for Authors

Dear authors, The manuscript titled “BIOLOGICAL CONTROL OF HYALOMMA TICKS IN CATTLE BY FUNGAL ISOLATES” is interesting, but several modifications are necessary before publication.

First, tick and fungus identification at the species level is mandatory. The authors may consider using a molecular approach for fungus identification.

Dear reviewer,

We appreciate you taking the time to thoroughly review our manuscript and provide constructive feedback. The point-by-point responses to the comments are as under.  

Response: As the ticks’ infestation is usually mixed infestation and not specie specific. Therefore, we used mixed population of Hyalomma spp.

Of course, it would be better if the fungi were identified at the species level. However, these fungal isolates will be identified up to the species level for our future study. Similarly, the metabolites obtained from these fungi will also be characterized for further investigations.

Literature shows that some researchers used fungal isolates against ticks, without their identification at species level, that the reason we identified the fungi just to the genus level, at least at this stage. “Several fungal species, such as Aspergillus, Beauveria, Fusarium, Paecilomyces, and Verticillium have been linked to ticks in the world (Estrada-Pea et al., 2014)”, Lines-86-88 of the revised manuscript.

  • Second, the fungal infection should be addressed. The authors can use histological sections stained with hematoxylin and eosin and molecular techniques (PCR andin situ hybridization).

Response: This is a valid point raised by the reviewer 4, but the histological and molecular studies were not included in our objectives. So, we didn’t apply these techniques and we will consider it for our future studies as per the suggestion of the reviewer.

  • Last but not least, despite being a promising biological control agent, the limitations of the entomopathogenic fungi extract should be better addressed in the discussion. For example, the possibility of side effects if the application is done directly on animals and its efficiency when the application is done in the environment and subject to environmental factors (e.g., temperature, rain, etc.)

Response: These points have been properly addressed and explained in the discussion part of the revised manuscript, as suggested by reviewer 4. Lines- 400-410.

Introduction:

Line 50: Add a space between “infection” and “Hyalomma spp.”

Response: Space added. Line-52

Line 50: Delete “spp.” after Hyalomma.

Response: spp. deleted. Line-52

There are plenty of scientific names written without italics. Please check through the text.

Response: All the scientific names have been made italics.

Line 67: Theileriasis is included in piroplasmosis.

Response: Theileriasis is deleted. Line-69.

Lines 70–72: “In the Kasur district of Pakistan, ticks were identified, blood protozoa were 70 detected by polymerase chain reaction testing in Friesian cattle, and blood parameters were 71 estimated (Hoogstraal, 1956)”. The paragraph is out of context.

Response: the sentence ‘ticks were identified, blood protozoa were detected by polymerase chain reaction testing in Friesian cattle and blood parameters were estimated (Hoogstraal, 1956)’ is deleted from the revised manuscript. The previous sentence is improved. Lines-70-72.

Lines: 87-90: “Ticks of the genus Hyalomma are dangerous ectoparasites that prey on animals, notably cattle. These ticks not only inflict direct harm on the host animals, but they also act as vectors for many infections, resulting in economic losses in the cattle business.”. This paragraph could be added at the end of line 64.

Response: Rectified as suggested by reviewer 4 in the revised manuscript. Lines-67-68.

Line 88: Change “prey” to “feed”.

Response: Changed. Line-90

 Materials and Methods:

The authors could add a map including the sites of sampling to assist the readers.

Response: Actually, the soil samples and the tick specimens were taken from only two sites of district Karak. That’s why the authors felt no need to provide the map.

Soil sample collection

How many samples were collected? What was the weight of each sample? When (period of the year) were the samples collected? Is there any variation during the period of the year (dry and wet seasons) that could influence the diversity of the fungi species in the soil? Please, clarify.

Response: Soil samples (10) were collected from 10 randomly selected places, during the months of July and August. The samples were pooled into a single sample, weighing about 2 Kg Lines- 126-128 of the revised manuscript.

Tick collection:

It is mandatory to identify the tick species analyzed during the study!

Response: As the tick’s infestation is generally a mixed infestation in animals that’s why the ticks selected for the experimental purpose were not identified up to species level.

In addition, the number of ticks in each experimental group must be added in this section.

Response: Five adult ticks and five tick’s larvae have been added. Rectified in the text of revised manuscript. Line 188-206

Isolation of fungi from the soil sample

Line 133: What does “MS” mean?

Response: MS tubes have been replaced with glass tubes. Line 135-137.

Line 136: What does “cc” mean?

Response: Replaced 1 cc with one ml in the revised manuscript. Line-140

Identification of the isolated fungi:

The authors could provide the molecular identification of the fungi at the species level. As well as the identification of ticks by level of species, this information will bring more clarity to the data obtained.

Response: Indeed, this is a very important point, and the fungi need to be identified at species level. Further studies have been recommended for the molecular identification of these fungal species at molecular level. Line 396-397 of the revised manuscript.

Adult, larvae and nymph mortality Assay:

Please add a comma after “Adult”

Response: The subheading Adult, larvae and nymph mortality Assay has been replaced with ‘Adult Mortality Assay’ in the revised manuscript. Line-186

Were the nymphs also tested?

Response: Nymphs were not tested in the current study, hence remove from the sub-heading. Line-186

Line 185: “five ticks”. Male or female? Engorged?

Response: ‘five ticks’- replaced with five adult engorged female ticks in the revised manuscript. Lines 188-189.

Results

Isolation and identification of fungi from the soil sample.

The authors must identify the fungus at the species level.

Response: As stated earlier, this must be done for further investigations as per the future recommendations. Lines-396-397

Line 230: “Figures 3.1, 3.2, and 3.3”. The figures may be improved. The letters (a or b) in the figures must be the same size and positioned at the same point, preferably in the top corner. Furthermore, the microscope image presents reflections. This must be corrected before publication. Were the photos taken with a cell phone?

Response: All the figures have been improved as per the valuable suggestion of the reviewer-4 in the revised manuscript.  Lines 242-258

Figures 4a and b are not necessary.

Response: Figures 4-a and have been removed from the revised manuscript as per suggestion of the reviewer4.

Figure 5 could be improved. For example, authors could present a more approximate picture. Moreover, the subtitles need to be more descriptive.

Response: The authors are also with the opinion that figure 5 is not necessary. Hence, the figure and its related description is hereby deleted from the revised manuscript.

Line 264: A maximum death rate of 5.0 was obtained, followed by 3.3” This rate is not clear to me. Could it be expressed in another way?

Response: ‘death rate’ was replaced with adult tick’s mortality to make it clear.  Lines- 265-274.

 Figures 6, 7, and 8 can be grouped into a single figure and distinguished from each other by the letters a, b, and c. Furthermore, the text referring to this data can be summarized.

Response: Of course, it would be appropriate to group all the figures in a single figure as per the suggestion of the reviewer 4. But while doing so, the format will be drastically changed and could not be accommodated in the current format. Moreover, to the authors, the text in the present condition is more understandable. So, the figures and the data were left unchanged in the revised manuscript.

Line 279: Figure 6. In the caption, it is mentioned “larval mortality” but in the topic (line 275), it’s talking about “adult ticks’ mortality”. Please, clarify.

Response: Figures 6, 7 and 8 are about the ‘larval mortality”. In the topics and in the respective text, it was rectified accordingly. Lines- 279-286.

Larval Mortality Assay against the fungal isolates

There was a statistical difference?

Response: The difference in the tick’s larval mortality due to different concentration of the fungal filtrate was significant. However, the graph (Figures 6, 7 and 8) shows the trend in the larval mortality due to the increase in fungal filtrates concentrations.

Dicussion:

Nowadays, more than 170 commercial products are based on entomopathogenic fungi formulations. However, the commercialization of these products still requires greater improvements to make them competitive against chemical pesticides. Thus, the authors should address these problems in this section. Please see the reference for more details. https://doi.org/10.1016/j.actatropica.2022.106627 

Response: Some more related and supportive literature has been added, as per the suggestion of the reviewer 4, that further elaborated the discussion part.

Furthermore, the authors must highlight the possible impacts of using this solution directly on animals, emphasizing the need to evaluate the side effects on cattle. Alternatively, the solution could be applied to the environment, and in this way, the efficiency of the product applied to the soil must be evaluated in the future in order to verify whether it remains viable (and for how long) when exposed to environmental variables (sun, rain, etc.).

Response: These valuable have been addressed in the revised manuscript like, in the in the controlled environment, and the mortality rates differed according to their filtrate concentrations. However, under natural conditions, the effect decreases due to the influenced by various environmental factors, such as temperature, relative humidity and exposure to solar radiation (Leemon et al., 2008, Fernandes et al., 2012). Therefore, it is necessary to evaluate the action of fungi under diverse environmental conditions.

The entomogenous fungi have several advantages over the chemical acaricides, as low environmental pollutant, low risk for animal health and high efficacy against pests (Barbieri et. al., 2023). So, a possible impact of these fungal filtrates on animals and their side effects on cattle need to be evaluated. Similarly, such fungi remain viable on cattle skin for several weeks (Kaaya et al., 1996), so the filtrates application at different intervals would be interesting protocols to explore in further studies. Lines 400-411 of the revised manuscript.

Lines 334-336: “For instance, M. anisopliae greatly reduced tick infestation in cattle. These fungi 334 use a variety of bioactive substances and processes to pierce the tick's cuticle, resulting in internal colonization and tick mortality.”. Please add a reference for this paragraph.

Response:

Line 340: Which tick species is it for?

Response: Hyalomma tick was used in the current study.

Lines 354-356: “Recent research has emphasized the biocontrol efficiency of 354 entomopathogenic fungi and the relevance of knowing their mode of action and molecular 355 interactions.”. Please add a reference for this paragraph.

Response: References have been provided, such as Bordin et al., 2022). Line- 368 and (Kaya et. el., 2011, Barbieri et. al., 2023) Line-370 of the revised manuscript.

Round 2

Reviewer 2 Report

Comments and Suggestions for Authors

Taking into account the limitations of the study presented by the authors, and that dramatically improved the caliad of the discussion

The Paper could be considered for publication